# Renieramycin T Inhibits Melanoma B16F10 Cell Metastasis and Invasion via Regulating Nrf2 and STAT3 Signaling Pathways

**DOI:** 10.3390/molecules27165337

**Published:** 2022-08-22

**Authors:** Baohua Yu, Jing Liang, Xiufang Li, Li Liu, Jing Yao, Xiaochuan Chen, Ruijiao Chen

**Affiliations:** 1Department of Pediatric Surgery, Affiliated Hospital of Jining Medical University, Jining 272067, China; 2College of Pharmacy, Heze University, Heze 274015, China; 3College of Basic Medicine, Jining Medical University, Jining 272067, China; 4Key Laboratory of Green Chemistry and Technology of Ministry of Education, College of Chemistry, Sichuan University, Chengdu 610064, China

**Keywords:** renieramycin T, tumor metastasis and invasion, p-STAT3, Nrf2

## Abstract

As one of marine tetrahydroisoquinoline alkaloids, renieramycin T plays a significant role in inhibiting tumor metastasis and invasion. However, the effect of renieramycin T on inflammation-related tumor metastasis and invasion is still unknown, and its mechanisms remain unclear. Here we established an inflammation-related tumor model by using the supernatant of RAW264.7 cells to simulate B16F10 mouse melanoma cells. The results indicate that renieramycin T suppressed RAW264.7 cell supernatant-reduced B16F10 cell adhesion to a fibronectin-coated substrate, migration, and invasion through the matrigel in a concentration-dependent manner. Moreover, Western blot results reveal that renieramycin T attenuated the phosphorylation of STAT3 and down-regulated the expression of Nrf2. Together, the above findings suggest a model of renieramycin T in suppressing B16F10 cancer cell migration and invasion. It may serve as a promising drug for the treatment of cancer metastasis.

## 1. Introduction

A malignant tumor has a higher metastatic capacity; the way of metastasizing includes tissue, blood and lymph, etc. [1]. Melanoma is a highly malignant skin tumor, which has various characteristics, such as high morbidity and mortality rate and early transfer [2,3,4,5,6]. Research shows that the most people die from metastatic carcinoma rather than primary carcinoma [1]. Therefore, restraining tumor migration and invasion may provide a valid strategy for tumor treatment.

The tumor microenvironment comprises a variety of nonmalignant stromal cells that play a pivotal role in tumor progression and metastasis [7]. Among them, tumor-associated macrophages (TAMs) are one of the most notable migratory hematopoietic cell types [8]. Clinical studies show that an increased number of TAMs frequently correlates with angiogenesis, metastasis, and poor prognosis [9]. Nevertheless, there is a consensus view that macrophage polarization is strongly related to tumor stage, suggesting that a dynamic switching from M1 phenotype, during the early phases of chronic inflammation, to an M2-like one might occur in established tumors. RAW264.7, a precursor of macrophage cell lines, was widely used in M1/M2 macrophage function research [10]. Moreover, RAW264.7 supernatant was proven to contain many inflammatory cytokines and chemokines, which were applied to establish an inflammation-related micro-environment [11].

As a transcription factor, NF-E2-relatedfactor 2 (Nrf2) mediate tumor progression via accelerated tumor proliferation [12,13,14]. It is verified that Nrf2 is up-regulated in lung, head, and neck squamous cell carcinoma tissues [15]. Further investigation shows that overexpression of Nrf2 enhances tumor chemo-resistance in some lung carcinoma, breast adenocarcinoma, and neuroblastoma cell lines [16]. It is reported that loss of Nrf2 in an oncogenic context-dependent manner can enhance cellular plasticity and motility, in part by using TGF-β/Smad signaling [17], and Nrf2 activation promotes lung cancer metastasis by inhibiting the degradation of Bach1 [18]. Nrf2 accumulation in lung cancers causes the stabilization of Bach1 by inducing HO-1, the enzyme catabolizing heme.

It is reported that aberrant STAT3 activation in tumor cells is associated with cell proliferation, cell survival, invasion, angiogenesis, and metastasis [19,20,21,22,23]. Conversely, targeting STAT3 activation inhibits tumor growth and metastasis both in vitro and in vivo without affecting normal cells, thus suggesting that STAT3 could be a valid molecular target for cancer therapy [24]. Moreover, STAT3 participates in tumor migration and invasion [25,26,27,28], and persistent activation of STAT3 in epithelial/tumor cells is linked to multiple human malignancies, including inflammation-associated cancer [29].

Marine tetrahydroisoquinoline alkaloids, are one kind of natural product that have a wide range of bioactivities, including anti-tumor, anti-bacteria, anti-virus, anti-inflammation activities, etc. [30]. Ecteinascidin-743 (ET-743) is the first marine antitumor drug to treat soft tissue sarcoma (STS) approved by the European Union in October 2007 [31]. Renieramycin T (RT), a bistetrahydroisoquinolinequinone that is extracted from the Thai blue sponge *Xestospongia* sp., for this study, was self-synthesized by Dr. Xiaochuan Chen from Sichuan University [32]. RT and its derivatives exhibited potent cytotoxicity against human lung cancer cells [33,34,35,36]. Nevertheless, the function of RT in mediating inflammation-related migration and invasion remains unclear.

In this study, we established an inflammation-related tumor model by using the supernatant of RAW264.7 cells to simulate B16F10 mouse melanoma cells. Moreover, we showed that RT suppressed RAW264.7 cell supernatant-reduced B16F10 cell adhesion to fibronectin-coated substrate, migration, and invasion through the matrigel in a concentration-dependent manner. Exploration of the underlying mechanism demonstrated that RT attenuated the phosphorylation of STAT3 and down-regulated the expression of Nrf2. The above findings suggest a model of RT in suppressing B16F10 cancer cell migration and invasion. It may serve as a promising drug for the treatment of cancer metastasis.

## 2. Results

### 2.1. RT Had No Effect on Inflammation-Induced B16F10 Cell Proliferation

Renieramycin T (RT), a bistetrahydroisoquinolinequinone that is extracted from the Thai blue sponge *Xestospongia* sp., in this study, was self-synthesized by our cooperative group. The structure of RT is shown in Figure 1A.

To investigate the role of RT in B16F10 cell proliferation, we treated B16F10 cells with various concentrations of RT, the activity of RT against B16F10 cells in a microenvironment was detected by CCK8 method. As shown in Figure 1B, after the control and conditioned cells were treated with RT, the viability of B16F10 cells induced by the supernatant of RAW264.7 was not changed by two concentrations of RT (80 ng/mL and 160 ng/mL). The results indicate that 0–160 ng/mL RT had no significant effect on inflammation-induced B16F10 cell proliferation.

### 2.2. RT Inhibits the Supernatant of RAW264.7 Cells-Induced B16F10 Cell Migration and Invasion

To explore the function of RT on B16F10 cell migration and invasion, we performed the wound healing assay and Transwell assay. Firstly, the migration ability of B16F10 was detected by scratch test. Compared with the control group, we found that the migration ability of B16F10 induced by RAW264.7 supernatant was enhanced, while the migration ability of B16F10 was reduced after the addition of RT treatment (Figure 2A,B). Correspondingly, the transwell results show that RT reversed the promoting effect of RAW264.7 supernatant on B16F10 invasion (Figure 2C,D). Collectively, these results indicate that RT inhibited the migration and invasion of B16F10 in an inflammatory microenvironment.

### 2.3. RT Inhibited the Expression of Metastasis-Related Genes in RAW264.7 Cells Supernatant-Induced Tumor Cells

To further verify the effect of RT on B16F10 metastasis, we detected the expression of tumor metastasis-related genes twist, snail, vimentin, and N-cadherin by qPCR. We found that RAW264.7 supernatant can enhance the expression of twist, snail, vimentin, and N-cadherin at the transcriptional level compared with the control group, but their expression decreased after RT treatment (Figure 3A–F). Moreover, the Western blotting results also show RAW264.7 supernatant increased the expression of twist, snail, vimentin, and N-cadherin, RT treatment reduced their expression (Figure 3G). The above results further indicate that RT inhibited the metastasis of B16F10 in an inflammatory microenvironment.

### 2.4. RT Suppressed the STAT3 and Nrf2 Signaling Pathways in B16F10 Cells

In order to explore the mechanism of RT against tumor invasion and metastasis, we detected the effects of RT on the expressions of STAT3, p-STAT3, and Nrf2 in B16F10 cells by Western blots and qPCR. The results show that RAW264.7 supernatant promotes the expression of STAT3, p-STAT3, and Nrf2 protein levels, and the expressions of STAT3, p-STAT3, and Nrf2 were recovered by RT treatment (Figure 4A–D). Similarly, at the mRNA level, RT was found to inhibit the attenuated phosphorylation of STAT3 and down-regulate the expression of Nrf2 induced by supernatant of RAW264.7 (Figure 4E,F). In conclusion, the above results display that RT may inhibit the migration and invasion of B16F10 in an inflammatory microenvironment through Nrf2 and STAT3 signaling pathways.

### 2.5. Proposed Model of RT in Suppressing B16F10 Cell Migration and Invasion

Based on the totality of our findings, we propose the following model (Figure 5). RT inhibits the migration and invasion of B16F10 treated with RAW264.7 supernatant through Nrf2 and STAT3 signaling pathways.

## 3. Discussion

As one of the major common human malignancies, melanoma has higher morbidity and mortality. As a result of tumor metastasis, clinical success in the surgical treatment of melanoma patients is limited. With the precision medicine developed, it is an urgent need to explore novel, new molecular agents for melanoma metastasis treatment.

Marine tetrahydroisoquinoline alkaloids are one kind of natural products that have a wide range of bioactivities, including anti-tumor, anti-bacteria, anti-virus, anti-inflammation activities, etc [30]. The most famous drug, Ecteinascidin-743 (ET-743), is the first marine antitumor drug to treat soft tissue sarcoma (STS) approved by the European Union in October 2007 [31]. Other small molecules of marine tetrahydroisoquinoline alkaloids, such as reieramycin M, were reported to kill the breast cancer cell MCF-7 by combining with doxorubicin to sensitize Anoikis’ resistance of lung cancer cell H460. As one of the marine tetrahydroisoquinoline alkaloids, RT plays a vital role in inhibiting tumor metastasis and invasion. In this study, the anti-metastasis ability of RT was discovered, and the mechanism of RT suppressing melanoma cell migration and invasion was reported for the first time.

Nuclear factor (erythroid-derived 2)-like 2 (Nrf2) is a redox-sensitive transcription factor that plays a key role in cellular cytoprotection against oxidative and electrophilic stress [37,38]. Although Nrf2 protects normal cells against chemically induced tumor formation, it confers an advantage for the survival and growth of many different types of cancer cells [39]. Another transcription factor, STAT3, functions as an oncogene. Persistently activated STAT3 is associated with the proliferation, survival, and invasiveness of tumor cells, including those of breast, lung, pancreatic, and head and neck origin. Recent studies suggested the possibility of the interplay or cross talk between Nrf2 and STAT3 [40,41]. In our studies, RT restrained the phosphorylation of STAT3 and decreased Nrf2 in protein levels, suggesting that the STAT3/Nrf-2 signaling axis might be involved in the mechanism of RT repressing migration and invasion. It is suggested that STAT3/Nrf2 can be an effective target for cancer prevention or treatment. However, the potential manner in which RT inhibits STAT3 and Nrf2 was not clear, and needs to be further studied.

## 4. Materials and Methods

### 4.1. Reagent and Antibody

RT was dissolved in dimethyl sulfoxide (DMSO) as a stock solution and stored at −20 °C until needed. The final concentration of DMSO did not exceed 0.1% throughout the study (this concentration was found to have no effect on cell invasion or cell growth). The chemical structure of RT is shown in Figure 1. The compound was prepared at concentrations of 80 ng/mL and 160 ng/mL. Primary antibodies for vimentin (CST, Boston, MA, USA), STAT3 (CST, Boston, MA, USA), p-STAT3 (CST, Boston, MA, USA) and Nrf2 (CST, Boston, MA, USA) were detected by Western blot. Antibody to GAPDH was from Santa Cruz Biotechnology (Santa Cruz, CA, USA). Cell Counting Kit-8 (CCK-8) was purchased from Good Laboratory Practice Bioscience (Glpbio, Montclair, CA, USA). IRDye TM800 conjugated second antibody was obtained from Rockland (Gilbertsville, PA, USA). BCA Protein Assay kit was purchased from Beyotime (Beyotime, Shanghai, China).

### 4.2. Cell Culture

The mouse melanoma cells B16F10 and the mononuclear cells RAW264.7 were originally obtained from the Cell Bank of Shanghai Institute of Cell Biology. B16F10 cells were cultured in DMEM medium (Gibco, Grand Island, NY, USA) containing 10% fetal bovine serum (Gibco, Grand Island, NY, USA), 100 U/mL penicillin, and 100 mg/L streptomycin. RAW264.7 cells were cultured in RPMI-1640 medium (Gibco, Grand Island, NY, USA) containing 10% heat-inactivated fetal bovine serum, 100 U/mL penicillin, and 100 mg/L streptomycin. Cells were grown in a stable environment with 5% CO_2_ at 37 °C.

### 4.3. Cell Viability Assay

B16F10 cells were plated at a density of 104 cells per well into 96-well plates. After overnight growth, cells were exposed to different concentrations of RT for 48 h in a 5% CO_2_ incubator at 37 °C. At the end of treatment, 10 μL of 0.5% CCK-8 was added to the medium and incubated for 4 h at 37 °C. The supernatant was removed and 0.1 mL DMSO was used to dissolve precipitate. Absorbance was measured spectrophotometrically at 450 nm.

### 4.4. Wound Healing Assay

B16F10 cells were seeded in a six-well plate and allowed to attach overnight to 80% confluency. Subsequently, cell monolayers were wounded by pipette tips and washed with PBS twice to remove floating cells. Cells were treated with or without different concentrations of RT for up to 48 h. Cells migrated into the wound surface and the number of migrating cells was determined under an inverted microscopy at various times; five randomly chosen fields were analyzed for each well. The percentage of inhibition was expressed using untreated wells at 100%. Three independent experiments were performed.

### 4.5. Transwell Assay

The ability of invasion was demonstrated by transwell assay. The melanoma B16F10 cells were treated with RAW264.7 supernate and different concentrations of RT solutions about 30 min in advance, and they were inoculated at a density of 1.5 × 104 cells in the upper chamber (Thermo Scientific, Shanghai, China), and the chemokine was in the lower chamber. The upper and lower chambers were separated by a polycarbonate membrane and a side near the upper chamber that was carpeted with Matrigel (Becton, Dickinson and Company, San Jose, CA, USA). Eventually, count the quantity of tumor cells.

### 4.6. Quantitative Real Time-PCR (qRT-PCR)

Total RNA was isolated from the cells using the Trizol reagent (Life Technology, Shanghai, China), and mRNA was reverse-transcribed to cDNA using the GoScript Reverse Transcription System (Vazyme, Nanjing, China). The qRT-PCR was performed in the Applied Biosystems 7500 cycler using the GoTaq qPCR Master Mix (Vazyme, Nanjing, China). The PCR conditions were as follows: pre-denaturation at 95 °C for 30 s, followed by 40 cycles of 95 °C for 10 s, and annealing and extension at 60 °C for 30 s. The relative cDNA levels of FR were calculated by the comparative Ct method, and normalized to GAPDH as the endogenous control (Table 1).

### 4.7. Western Blot

Cells were treated with RAW264.7 supernate and various concentrations of RT for 24 h. Cells were lysed with RIPA buffer, and the mixed protein was separated by SDS-PAGE according to the size of molecule, and transferred to the NC membrane. The membranes were combined with primary antibody to GAPDH, STAT3, p-STAT3, and Nrf2, respectively. They were co-incubated in the room temperature for 2 h. After washing with TBST three times, the membranes were combined by secondary antibody for 1 h at room temperature. The blots were detected with enhanced chemiluminescence (Pierce Chemical, 34080, Rockford, IL, USA) and then analyzed by ImageJ software (National Institutes of Health, Bethesda, MD, USA).

### 4.8. Statistical Analysis

All the data in our results are shown as means ± standard deviation (SD) from triplicate experiments. Statistically significant differences [analysis of variance (ANOVA) and post hoc tests] were analyzed using the GraphPad Prism software (GraphPad Software Inc., Avenida, CA, USA). Details of each statistical analysis used are provided in the figure legends.

## 5. Conclusions

In conclusion, we established an inflammation-related tumor model by using the supernatant of RAW264.7 cells to simulate B16F10 mouse melanoma cells. We showed that RT suppressed RAW264.7 cell supernatant-reduced B16F10 cell adhesion to fibronectin-coated substrate, migration, and invasion through the matrigel in a concentration-dependent manner. We explored the underlying mechanism and demonstrated that RT attenuated the phosphorylation of STAT3 and down-regulated the expression of Nrf2. The above findings suggest a model of RT for suppressing B16F10 cancer cell migration and invasion. It may serve as a promising drug for the treatment of cancer metastasis.

## Figures and Tables

**Figure 1 molecules-27-05337-f001:**
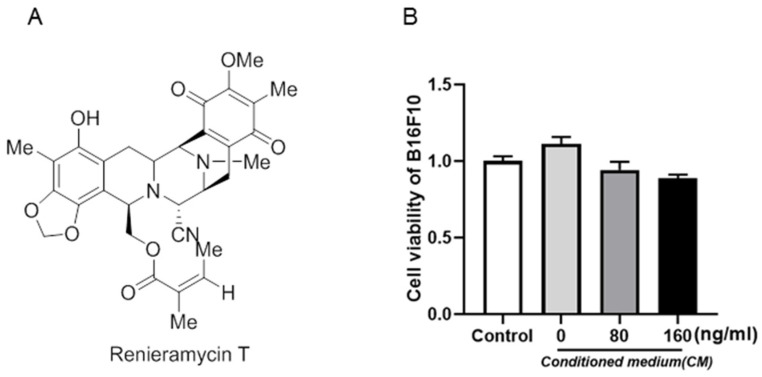
The structure and proliferation effects of renieramycin T (RT) were shown. (**A**) The structure of renieramycin T (RT) is shown. (**B**) RT had no effect on cell proliferation of B16F10 cells in the microenvironment.

**Figure 2 molecules-27-05337-f002:**
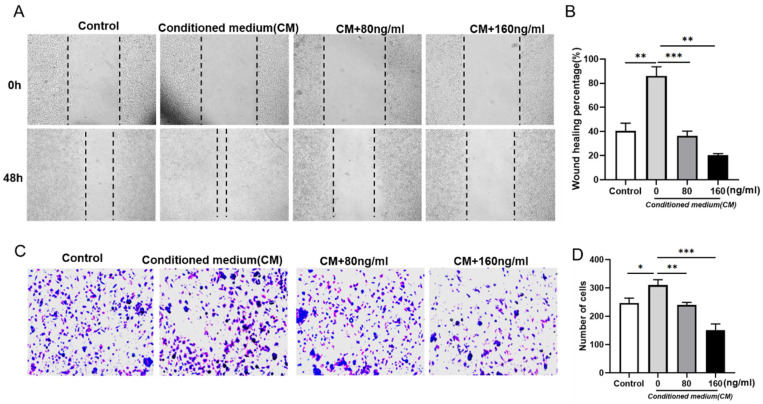
Effect of RT on B16F10 migration in tumor microenvironment. (**A**) Images of cell migration in each group within 48 h of scratch in wound healing experiment; (**B**) Statistics of wound healing area of B16F10 cells (** *p* < 0.01, *** *p* < 0.001). (**C**) Images of groups of cells passing through pores in the Transwell experiment; (**D**) Statistics on the number of B16F10 invaded cells (* *p* < 0.05, ** *p* < 0.01, *** *p* < 0.001).

**Figure 3 molecules-27-05337-f003:**
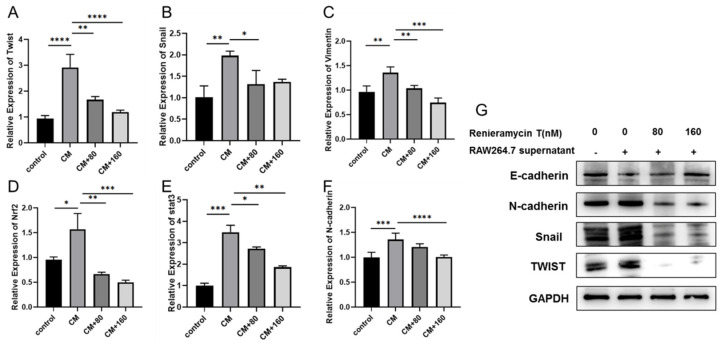
RT inhibits the expression of B16F10 tumor metastasis-related genes. (**A**–**F**) Quantitative analyses of twist, snail, vimentin, and N-cadherin mRNA in B16F10 treated with RT for 48 h at the control and conditioned cells. (**G**) Western blotting was performed to check the expression of twist, snail, vimentin, and N-cadherin in B16F10 treated with RT for 48 h at the control and conditioned cells. (n = 3, * *p* < 0.05, ** *p* < 0.01, *** *p* < 0.001, **** *p* < 0.0001).

**Figure 4 molecules-27-05337-f004:**
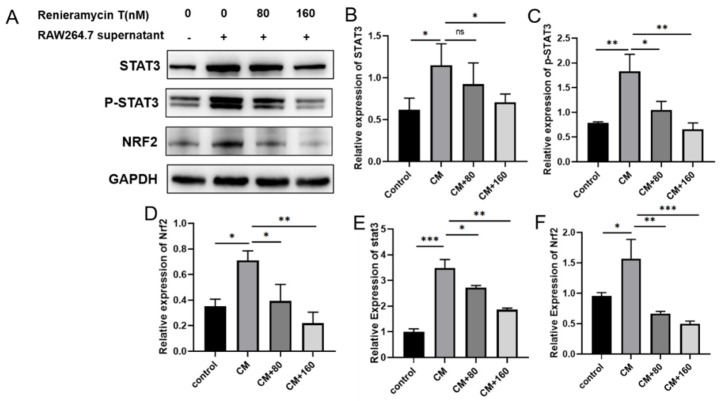
RT repressed the phosphorylation of STAT3 and down-regulated the expression Nrf2 in B16F10 cells. (**A**) The expression of STAT3, p-STAT3, and Nrf2 proteins were detected by WB assay; (**B**–**D**) The relative expression level of p-STAT3, STAT3, and Nrf2 proteins were analyzed in the B16F10 (* *p* < 0.05, ** *p* < 0.01); (**E**,**F**) The relative expression of STAT3 and Nrf2 were detected by qRT-PCR (* *p* < 0.05, ** *p* < 0.01, *** *p* < 0.001).

**Figure 5 molecules-27-05337-f005:**
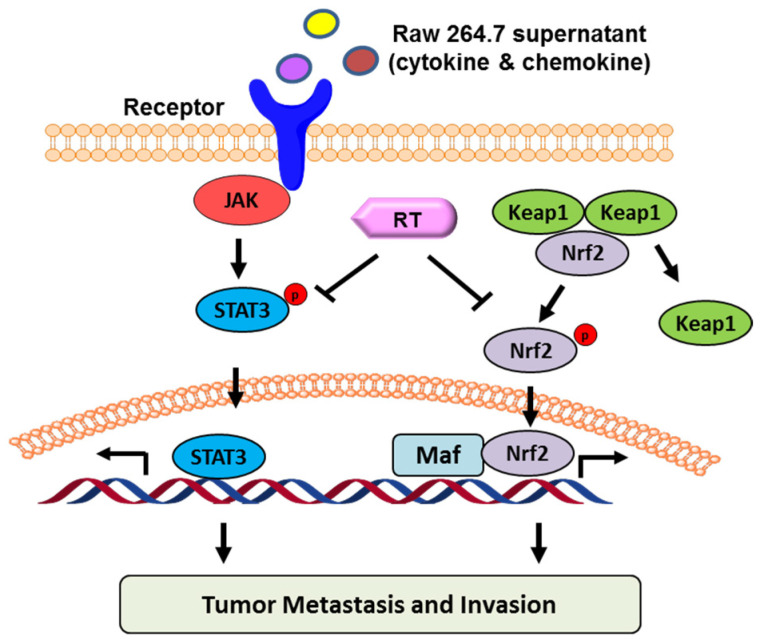
Proposed model of RT suppresses B16F10 cell migration and invasion.

**Table 1 molecules-27-05337-t001:** Real-time RT-PCR oligonucleotide primer.

Genes	Primer	Sequence (5′-3′)
GAPDH	Forward Reverse	GCACCGTCAAGGCTGAGAAC TGGTGAAGACGCCAGTGGA
Twist	Forward Reverse	GTCCGCAGTCTTACGAGGAG GCTTGAGGGTCTGAATCTTGCT
Snail	Forward Reverse	TCGGAAGCCTAACTACAGCGA AGATGAGCATTGGCAGCGAG
Vimentin	Forward Reverse	GACGCCATCAACACCGAGTT CTTTGTCGTTGGTTAGCTGGT
N-cadherin	Forward Reverse	TCAGGCGTCTGTAGAGGCTT ATGCACATCCTTCGATAAGACTG
STAT3	Forward Reverse	CAGCAGCTTGACACACGGTA AAACACCAAAGTGGCATGTGA
Nrf2	Forward Reverse	TCAGCGACGGAAAGAGTATGA CCACTGGTTTCTGACTGGATGT

## Data Availability

Data are available from the authors upon request.

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
