# Peer review of "Renieramycin T Inhibits Melanoma B16F10 Cell Metastasis and Invasion via Regulating Nrf2 and STAT3 Signaling Pathways"

_molecules, 2022, doi:10.3390/molecules27165337_

Round 1

Reviewer 1 Report

In this manuscript, the authors reported that renieramycin T inhibited melanoma B16F10 cell metastasis and invasion via regulating Nrf2 and STAT3 signaling pathways. The methods are sound. Several major points may need to be addressed:

1. Uses of the terms: metastasis, migration, and invasion are confusing. It is recommended to clarify their relationship. 

2. It is recommended to study the protein expression of Twist, Snail, Vimentin, and N-cadherin after RT treatment by western blotting assay.

3. There are several errors in formatting. E.g. In the introduction, the spaces before the reference numbers are inconsistent, double periods, and the italics are not used correctly. 

It is also recommended to double-check and standardize the format of references. For example, the 21, 26-29 references have no page number.

4. The authors did not study MMP2 and MMP9 in this work, these two genes may not need to be shown in Figure 6. The author has also repeatedly emphasized that the drug affects the tumor microenvironment. The authors may like to provide experimental evidence to support this statement.

5. The authors reported aberrant STAT3 activation in tumors is associated with cell proliferation, invasion, and metastasis. Inhibition of STAT3 activation is effective in cancer treatment. It is recommended to provide some examples in the introduction. The following papers may be useful.

1) Proceedings of the National Academy of Sciences, 2005, 102(17): 5998-6003.

2) Cancer letters, 2017, 396: 76-84.

3) Investigational new drugs, 2012, 30(3): 916-926.

4) Angewandte Chemie, 2014, 126(35): 9332-9336.

Overall, I recommend the publication of this manuscript after major revision.

Author Response

Dear Reviewer #1:

We very much appreciated your constructive critiques and comments regarding our manuscript, which we have now completely revised according to your suggestions. We performed some additional experiments to address your concerns. Many critical changes have been made which are highlighted in red and included in this newly revised version. We offer here our point-by-point response to your questions and concerns:

Point 1: Uses of the terms: metastasis, migration, and invasion are confusing. It is recommended to clarify their relationship.

Response 1: Thanks for your suggestion. As far as we know, migration and invasion mean tumor cell short distances movement, the detecting methods of migration were wound healing and trans-well, the detecting method of invasion often use trans-well combined with matrigel in the in vitro experiments. However, metastasis means tumor cell long distances movement in vivo. In our experiments, we detected cell migration and invasion in figure 3.

Point 2:  It is recommended to study the protein expression of Twist, Snail, Vimentin, and N-cadherin after RT treatment by western blotting assay.

Response 2: Following your suggestion, we detected the expression of Twist, Snail, Vimentin, and N-cadherin after RT treatment by western blotting.  We added it in figure 3G of the revised manuscript.

Point 3:  There are several errors in formatting. E.g. In the introduction, the spaces before the reference numbers are inconsistent, double periods, and the italics are not used correctly.

It is also recommended to double-check and standardize the format of references. For example, the 21, 26-29 references have no page number.

Response 3: Following your suggestion, we have revised it and double-checked the format. We also added the page number of references 21, 26-29.

Point 4:  The authors did not study MMP2 and MMP9 in this work, these two genes may not need to be shown in Figure 6. The author has also repeatedly emphasized that the drug affects the tumor microenvironment. The authors may like to provide experimental evidence to support this statement.

Response 4: Following your suggestion, we removed MMP2 and MMP9 in Figure 5 of new version manuscript.

Point 5:  The authors reported aberrant STAT3 activation in tumors is associated with cell proliferation, invasion, and metastasis. Inhibition of STAT3 activation is effective in cancer treatment. It is recommended to provide some examples in the introduction. The following papers may be useful.

Response 5: Following your suggestion, we have cited the references.

Sincerely,

Ruijiao Chen

Reviewer 2 Report

Manuscript entitled “Renieramycin T inhibits melanoma B16F10 cell metastasis and invasion via regulating Nrf2 and STAT3 signaling pathways” by Baohua Yu et al. describes that marine alkaloid, renieramycin T can inhibit tumor (melanoma B1610) metastasis and invasion. It can be considered as a drug against cancer metastasis.

Although the work sounds scientific, unfortunately it is written very casually. Fig. 2 is described in the text as Fig. 1. On the other hand, about Fig. 1 there is no word. Moreover, the inflammation related tumor model is very insufficiently described. The introduction should be more informative. The authors did not write it just for themselves!

They found that RT inhibits the supernatant of RASW264.7 cell - induced B1610 migration and invasion. Next, a working model is proposed how RT suppress melanoma B1610 cells migration and invasion. IL-6 (!?) has suddenly been proposed as the main trigger of the process that RT is supposed to inhibit. 

Manuscript submitted to a scientific journal should be written to an obvious standard. This one is written very carelessly. In addition, this text has many problems with spaces, which also underlines the lack of due diligence in its preparation.

Author Response

Dear Reviewer #2:

We very much appreciated your constructive critiques and comments regarding our manuscript, which we have now completely revised according to your suggestions. We performed some additional experiments to address your concerns. Many critical changes have been made which are highlighted in red and included in this newly revised version. We offer here our point-by-point response to your questions and concerns:

Point 1: Although the work sounds scientific, unfortunately it is written very casually. Fig. 2 is described in the text as Fig. 1. On the other hand, about Fig. 1 there is no word. Moreover, the inflammation related tumor model is very insufficiently described. The introduction should be more informative.

Response 1:We are very sorry for our casually writing. We have revised figure 1 & 2 in the revised manuscript. We also added the description of inflammation related tumor model in the introduction.

Point 2: They found that RT inhibits the supernatant of RASW264.7 cell - induced B1610 migration and invasion. Next, a working model is proposed how RT suppress melanoma B1610 cells migration and invasion. IL-6 (!?) has suddenly been proposed as the main trigger of the process that RT is supposed to inhibit.

Response 2: We are very sorry for this. We checked the STAT3 activation and supposed IL-6 was the up-stream of STAT3. We noticed that it was inaccurate. We have revised this in the new version manuscript.

Point 3:Manuscript submitted to a scientific journal should be written to an obvious standard. This one is written very carelessly. In addition, this text has many problems with spaces, which also underlines the lack of due diligence in its preparation.

Response 3: We are very sorry for our casually writing and format. We have now completely checked and revised according to the reviewers’ suggestions.

Sincerely,

Ruijiao Chen

Round 2

Reviewer 1 Report

The authors have satisfactorily responded to all the questions and made the necessary changes to the manuscript. I have no further questions and suggest the acceptance of the revised manuscript.

Reviewer 2 Report

Accept in present form